



# The sensitivity of Alpine summer convection to surrogate climate change: An intercomparison between convection-parameterizing and convection-resolving models

Michael Keller[1,2], Oliver Fuhrer[3], Nico Kröner[1], Daniel Lüthi[1], Juerg Schmidli[1,4], Martin Stengel[5], Reto Stöckli[3], and Christoph Schär[1]

[1]Institute for Atmospheric and Climate Science, ETH Zürich, Zurich, Switzerland
[2]Center for Climate Systems Modeling (C2SM), ETH Zürich, Zurich, Switzerland
[3]Federal Office of Meteorology and Climatology MeteoSwiss, Zurich, Switzerland
[4]Institute for Atmospheric and Environmental Sciences, Goethe University, Frankfurt am Main, Germany
[5]Deutscher Wetterdienst (DWD), Offenbach, Germany

*Correspondence to:* M. Keller (michael.keller@env.ethz.ch)

**Abstract.** Climate models project an increase in heavy precipitation events in response to greenhouse gas forcing. Important elements of such events are rain showers and thunderstorms, which are poorly represented in models with parameterized convection. In this study, simulations with 12 km horizontal grid spacing (convection-parameterizing model, CPM) and 2 km grid spacing (convection-resolving model, CRM), and with either a one-moment microphysics scheme (1M) or a two-moment microphysics scheme (2M) are employed to investigate the change in the diurnal cycle of convection with warmer climate. For this purpose, simulations of 11 days in June 2007 with a pronounced diurnal cycle of convection are compared with surrogate simulations from the same period. The surrogate climate simulations mimic a future climate with increased temperatures, but unchanged relative humidity and synoptic-scale circulation. Two temperature scenarios are compared, one with homogeneous warming (HW) using a vertically uniform warming, the other with vertically-dependent warming (VW) that enables changes in lapse rate.

The two sets of simulations with parameterized and explicit convection exhibit substantial differences, which are well known from the literature. These include differences in the timing and amplitude of the diurnal cycle of convection, and the frequency of precipitation with low intensities. There are also significant differences in terms of the response to the surrogate warming. For CRM, an increase of hourly heavy precipitation events is found for both surrogate scenarios and microphysics schemes. The intensification is consistent with the Clausius-Clapeyron relation. For cloud type frequencies, virtually no changes are found for HW, but a substantial reduction in high clouds is found for VW. Some of the CPM sensitivities differ significantly. Importantly, the increase of heavy precipitation events simulated by CPM is larger than suggested by the Clausius-Clapeyron relation. Moreover, significant differences between CPM and CRM are found in terms of the radiative feedbacks, with CRM exhibiting a stronger negative feedback in the top of the atmosphere energy budget.



# 1 Introduction

The diurnal cycle of convective clouds and precipitation over Europe is mainly active during summer, when solar radiation is strongest. The available energy at the Earth's surface is partitioned into sensible and latent heat fluxes, which in turn are redistributed in the atmosphere by convective processes. If the resulting updrafts are strong enough and persistent, this leads to

high cloud tops, which can be detected as cold temperatures in satellite measurements. In these, the diurnal cycle of summertime convection over Europe is found to be strongest over mountain areas, such as the Alps (Levizzani et al., 2010). A more conventional indicator for deep convection is surface precipitation. In line with the satellite measurements, pronounced seasonal maxima are found in summer along the Alpine ridge (e.g. Frei and Schär, 1998).

The diurnal cycle of summertime convection has been investigated by conventional convection-parameterizing models

(CPMs) and high-resolution convection-resolving models (CRMs). Both approaches have specific advantages. Long-term global climate projections need significantly more computer resources than weather forecasts of a few days. Thus, climate simulations are typically conducted using CPMs. The CPMs lack a good representation of the diurnal cycle of convection (Bechtold et al., 2004; Brockhaus et al., 2008; Hohenegger et al., 2008), which is improved in CRMs (Schlemmer et al., 2011; Langhans et al., 2013; Prein et al., 2013). In addition to the improvement in the diurnal cycle, improvements were also found in

the frequencies of wet days and heavy precipitation events (Ban et al., 2014). In recent years, it has become possible to conduct decade-long CRM climate projections on regional (Kendon et al., 2014; Ban et al., 2015) and continental scales (Leutwyler et al., 2016, 2017). A review on climate simulations with CRMs can be found in Prein et al. (2015).

Projections of the summer climate over central Europe have found an increase in daily heavy precipitation events despite reductions in mean precipitation amounts (Christensen and Christensen, 2003; Frei et al., 2006; Rajczak et al., 2013). An

intensity increase is also found for hourly heavy precipitation events, both by CPM and CRM simulations (Ban et al., 2015). Past research has indicated that changes in precipitation extremes are limited by the water vapor content in a warmer climate. This limitation follows the Clausius-Clapeyron relation (6–7 % K$^{-1}$) (e.g. Allen and Ingram, 2002). This argumentation is supported for daily events in a number of studies. However, for hourly events some studies project an increase beyond this relation (Lenderink and Van Meijgaard, 2008; Kendon et al., 2014), while other studies confirm the Clausius-Caperyron scaling

(Ban et al., 2015). Therefore, further investigations of the changes in precipitation extremes are needed.

The expected changes in precipitation climate will be governed by a number of factors. To distinguish between thermodynamic and circulation contributions, surrogate experiments can be conducted (Schär et al., 1996). In these regional climate model (RCM) experiments, the temperature distributions at the lateral boundaries are changed consistent with the expected large-scale warming, but relative humidity and circulation are held constant. Experiments of this type have revealed signif-

icant changes in mean precipitation and precipitation statistics when applying a vertically homogenous warming (HW) for mid-latitude conditions (Frei et al., 1998; Seneviratne et al., 2002; Im et al., 2010; Attema et al., 2014). However, climate change studies also show that there are pronounced stratification changes. More specifically, the upper troposphere is projected to warm at a faster rate than the surface (Santer et al., 2008; Collins et al., 2013). This implies that a vertically-dependent warming (VW) is closer to what is expected for the future. Kröner et al. (2016) found that the associated stratification (or lapse





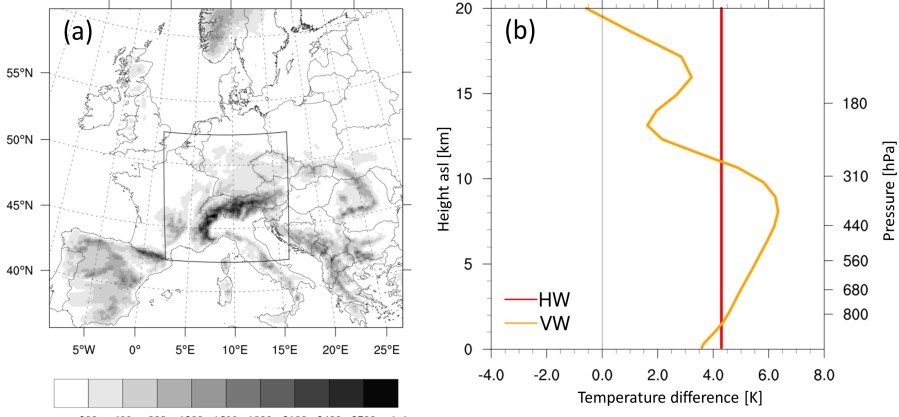

**Figure 1.** (a) Computational domains of the CPM simulations (full domain, 12 km resolution) and the CRM simulations (box in the center, 2 km resolution). Topography (m) is indicated in gray shading. The box also corresponds to the analysis domain. (b) Height profiles of the temperature differences for homogeneous warming (HW) and vertically-dependent warming (VW) relative to the control at the lateral boundaries of the CPM simulation domain, averaged over the investigated period (taken from Kröner et al. (2016)). Height is indicated in km on the left side, and pressure values at particular heights, averaged over the 11-day period, are indicated in hPa on the right side.

rate) effect explains one third of the projected changes in north-south temperature gradient of the European summer climate. Furthermore, they showed that the stratification changes strongly modulate convective precipitation. In the current study, we will use a related methodology and address stratification effects in the framework of CPM and CRM simulations.

For completeness it should be mentioned that a further surrogate approach exists, which is called pseudo-global warming

(e.g. Rasmussen et al., 2011; Prein et al., 2016). There, the main difference to VW is that the temperature change is not only a function of height but also of the spatial coordinates. To keep atmospheric dynamics in balance, this implies that also the climate change signal of other variables has to be calculated explicitly.

In Keller et al. (2016), an 11-day period in June 2007 with a pronounced diurnal cycle of convection is investigated by evaluating CPM and CRM simulations with satellite data. One of the major outcomes of that paper is that using a two-

moment microphysics scheme (2M) with ice sedimentation, instead of the standard one-moment microphysics scheme (1M) without ice sedimentation, reduces the high cloud cover bias, but without significantly affecting the precipitation response. The current paper builds on the previous study and expands it with surrogate simulations (HW and VW) for the same period. Apart from a small change in the setup (see Sect. 2.1), the control simulations are identical to the simulations in Keller et al. (2016). We address the following three questions: How will the diurnal cycle of convection and the associated precipitation and

clouds change in a warmer climate? How large is the impact of different temperature change profiles (HW versus VW)? How do the simulated changes depend on the modeling framework, in particular on the horizontal resolution (CPM versus CRM simulations) and the cloud-microphysics parameterization (1M versus 2M schemes)?





The paper is structured as follows: In Sect. 2, the COSMO setup, the surrogate setup, the analysis methodology, and the observations used for evaluation are introduced. The results are presented and discussed in Sect. 3, and finally, the conclusions are presented in Sect. 4.

## 2 Methods and data

### 2.1 Model and surrogate setup

This study uses the COSMO model (Consortium for Small-Scale Modeling) in climate mode (referred to as COSMO-CLM) at kilometer-scale resolution (Baldauf et al., 2011). The setup is close to previous studies (e.g. Ban et al., 2014; Keller et al., 2016) and convection-resolving simulations in numerical weather prediction (NWP) mode at MeteoSwiss (e.g. Weusthoff et al., 2010). For this study, CPM simulations (at a grid spacing of 12 km) and CRM simulations (at a grid spacing of 2.2 km) are conducted, following the setup of Keller et al. (2016). The CPM simulations are conducted over Europe and initialized and driven by ERA interim, with the exception of initial soil moisture conditions, which are taken from a ten-year climate run of Ban et al. (2014). The CRM simulations are conducted over an extended Alpine area and are initialized and driven by the CPM simulations. All CPM simulations use a one-moment microphysics scheme (1M) (Reinhardt and Seifert, 2006), while for the CRM simulations both the 1M or a two-moment microphysics scheme (2M) (Seifert and Beheng, 2006) are employed. The only significant difference to the setup of Keller et al. (2016) is that both CRM and CPM simulations use the same root depth. The root depth defines the lowest level from which plants can take water and use for transpiration (Doms et al., 2011). The analysis is performed over the CRM domain (Fig. 1a) for all simulations. Further information about the setup used for the 2M can be found in Keller (2016, Sect. 2.1.1).

In addition to the control (CTRL) simulations mentioned above, six surrogate simulations are conducted. Within these surrogate simulations, two different ways of surrogate warming are applied: a homogeneous warming (HW) and a vertically-dependent warming (VW) (Fig. 1b). The specifications of all simulations are summarized in Table 1. Schär et al. (1996) showed that for a pressure-dependent but spatially independent temperature change $\Delta T(p)$, the same flow fields satisfy the hydrostatic set of governing equations. As the model levels of COSMO are not expressed in pressure coordinates, a height-dependent change $\Delta T(z)$ is specified for simplicity, but the resulting change in the mass balance is negligible.

In applying the methodology, we follow Kröner et al. (2016): For calculating the temperature difference profiles $\Delta T$ of HW and $\Delta T(z)$ of VW, one of the core simulations of the CMIP5 project (Taylor et al., 2012) was used. The simulation follows the RCP 8.5 scenario (representative concentration pathways) (Moss et al., 2010). This scenario represents a relatively high greenhouse gas emissions pathway with an expected radiative forcing of 8.5 W m$^{-2}$ at the end of the century (Riahi et al., 2011). This high emission scenario was chosen for this study to amplify potential differences between present and future climates. The simulation chosen for this study is from the Max Planck Institute (MPI). It was calculated with an earth system model (ESM), which couples an atmospheric model with an ocean model and a vegetation model. The atmospheric part of the model is the ECHAM6 model (Stevens et al., 2013), which includes a carbon cycle model, and has a "low" vertical resolution (LR, 47 layers). The full model is called MPI-ESM-LR (Giorgetta et al., 2013). The HW and VW profiles were calculated





**Table 1.** Overview and specifications of the simulations analyzed in this paper.

| Name | Spatial resolution | Microphysics scheme | Convection scheme | Initial and boundary conditions | Initial date | Domain (see Fig. 1a) |
|---|---|---|---|---|---|---|
| CTRL_12km1M | 12 km | one-moment (1M) | shallow and deep | ERA interim[a] | 1 Oct 2006, 00 UTC | Europe |
| CTRL_2km1M | 2.2 km | one-moment (1M) | shallow | CTRL_12km1M | 1 Apr 2007, 00 UTC | Alpine region |
| CTRL_2km2M | 2.2 km | two-moment (2M) | shallow | CTRL_12km1M | 1 Apr 2007, 00 UTC | Alpine region |
| HW_12km1M | 12 km | one-moment (1M) | shallow and deep | ERA interim[a] + HW | 1 Oct 2006, 00 UTC | Europe |
| HW_2km1M | 2.2 km | one-moment (1M) | shallow | HW_12km1M | 1 Apr 2007, 00 UTC | Alpine region |
| HW_2km2M | 2.2 km | two-moment (2M) | shallow | HW_12km1M | 1 Apr 2007, 00 UTC | Alpine region |
| VW_12km1M | 12 km | one-moment (1M) | shallow and deep | ERA interim[a] + VW | 1 Oct 2006, 00 UTC | Europe |
| VW_2km1M | 2.2 km | one-moment (1M) | shallow | VW_12km1M | 1 Apr 2007, 00 UTC | Alpine region |
| VW_2km2M | 2.2 km | two-moment (2M) | shallow | VW_12km1M | 1 Apr 2007, 00 UTC | Alpine region |

[a]Soil moisture for initial conditions is from a ten-year climate run of Ban et al. (2014).

by masking the EURO-CORDEX domain (Jacob et al., 2014) in the MPI-ESM-LR simulation (ensemble member r1i1p1). This domain is slightly larger than the area of the 12 km simulations of this study. For the profiles, a mean annual cycle of the difference between the spatially averaged 30-year means of 1971 to 2000 and 2070 to 2099 was taken. This annual cycle was smoothed following the spectral smoothing method of Bosshard et al. (2011). The resulting time- and height-dependent profile was taken for VW. For the profile of HW, the temperature values at 850 hPa were applied over the full height. The SST change signal is equal to the $\Delta T$ change signal of the lowest atmospheric level, which neglects a possible change in the land-sea temperature contrast.

For comparison to the observations, cloud top pressure (CTP) and cloud optical thickness (COT) are calculated after the methodology used in Keller et al. (2016). Outgoing longwave radiation (OLR) and reflected solar radiation (RSR) are standard outputs of COSMO.



## 2.2 Calculation of scaling rate

In Section 3.2, the scaling rate ($SR$) for different percentiles ($p$) is calculated as:

$$SR^s(p) = \frac{P^s(p) - P^c(p)}{P^c(p) \cdot \Delta T} \tag{1}$$

where $P$ is hourly accumulated precipitation, $s$ the surrogate warming simulation, $c$ the corresponding control simulation, and $\Delta T$ the spatially and temporally averaged change in 2 m temperature between $s$ and $c$.

The percentiles are calculated using both wet and dry events following Ban et al. (2015) and Schär et al. (2016). Before calculating the percentiles, data is pooled over the full analysis domain. This step differs from the method used in Ban et al. (2015). In their study, the calculation of the percentiles and the normalization by temperature was done for every grid point before averaging. With their method, the statistics are calculated for all grid points, independently of their spatial climatology, while with pooling the data of some region is treated as one sample. The method with pooling was chosen in this study, due to the small dataset (11-day period).

## 2.3 Observations

*Precipitation data*

For this study, the gridded precipitation dataset EURO4M-APGD (Isotta and Frei, 2013) is employed, which is based on rain-gauge measurements across the European Alps and adjacent areas (Isotta et al., 2014). EURO4M is a collaborative project of the European Union. The dataset has a daily temporal and 5 km spatial resolution. Known limitations of this product are an underestimation of high precipitation intensities and an overestimation of low precipitation intensities (Isotta et al., 2014).

*Satellite data*

Cloud properties of the Cloud_cci MODIS-Aqua dataset (Stengel et al., 2017) are used in our study, i.e. Level-3U data which contains unaveraged, pixel-based retrievals sampled on a regular longitude-latitude grid with a resolution of 0.02° covering Europe. The scientific content of these data is described in Stengel (in prep.). Cloud variables used in our study are cloud top pressure (CTP) and cloud optical thickness (COT). As the actual cloud detection value (cloudy or clear) comes with an uncertainty estimate on pixel level, we used the latter to only collect CTP and COT for pixels with a low uncertainty in cloud detection. We rejected all cloudy pixels for which the detection uncertainty exceeded 35 %. This value is somewhat arbitrary but mainly based on analysing the relative frequency of cloud detection uncertainty which yielded in a bimodal distribution when including all cloudy pixels, with 35 % being approximately the value separating the more certain from the more uncertain clouds. It needs to be noted that the omitted, more uncertain cloudy pixels are associated with mostly high CTPs, thus low-level clouds. This potentially biases the used satellite data and needs to be kept in mind for the comparison later on. All satellite data outside the analysis domain (Fig. 1a) is omitted. Model equivalent CTP and COT values are selected from model time steps at 13 UTC to match the Aqua satellite overpass time of approximately 1:30 pm (local time).





The Geostationary Earth Radiation Budget (GERB) radiometer is onboard the Meteosat Second Generation (MSG) satellites. These satellites are geostationary, enabling a temporal resolution of 15 min. Outgoing longwave radiation (OLR) and reflected solar radiation (RSR) data are used in this study, which was produced at the Royal Meteorological Institute of Belgium (RMIB) (Dewitte et al., 2008) after the methodology of Harries et al. (2005), who state an error of $< 1$ % for both products. The

spatial resolution of 9x9 km$^2$ at the sub-satellite point at the equator becomes approximately 12x18 km$^2$ over the Alps (cf. EUMETSAT, 2013).

## 3   Results

Our study period is from 3 to 13 June 2007, which was characterized by a pronounced diurnal cycle of convection over the Alps and surrounding areas with a maximum of precipitation and high cloud cover in the afternoon (Keller et al., 2016). This

synoptic situation makes the time period ideal to study the diurnal cycle of convection under relatively undisturbed conditions. The changes in this diurnal cycle due to surrogate climate change are investigated in this section. First, differences between the introduced temperature profiles at the lateral boundaries (Fig. 1b) and the resulting profiles inside the domain are studied. Second, the impact of the surrogate warming on precipitation and clouds is investigated.

We are aware that an 11-day period is very short from a climatological perspective. But we consider this as a process-oriented

study. Here, important parameters such as the large-scale flow are constrained, therefore the internal variability becomes smaller than in classical climate studies and shorter periods can be investigated.

### 3.1   Vertical temperature and humidity profiles

Figs. 2a and b show the differences of HW and VW with respect to control (CTRL) of the spatially and temporally averaged temperature profiles inside the analysis domain from hourly data during the 11-day period. Differences are taken between the

12km1M runs (Fig. 2a), the 2km1M runs (Figs. 2a, b), and the 2km2M runs (Fig. 2b). The differences imposed at the CPM-boundaries are indicated in dark green (HW) and light green (VW). Overall, the profiles of all three HW runs and of all three VW runs are comparable, with the smallest differences due to different microphysics schemes. This indicates similar surrogate conditions for all HW and all VW simulations. Below 4.5 km, the profiles of HW and VW are quite similar, despite the large differences in the initial profiles. From 4.5 to 11 km, the difference between HW and VW increases, and for all profiles an

enhanced warming with height is found. Above 11 km, the profiles approximate the lateral boundary profiles (green) with height, since temperature is relaxed to the driving model toward the upper boundary. In comparison to the lateral boundary profiles, a cooling is found below 6.5 km for HW and below 8.5 km for VW. Above these heights and below 12 km, the atmosphere becomes warmer than the originally introduced warming. We assume that the redistribution of temperatures in the model domain compared to the boundaries is caused by convection. The transformation of the initial HW-profile (dark green)

to profiles (red) that are similar to the initial VW-profile (light green) indicates that the VW experiment is closer to the model equilibrium than HW for this period.





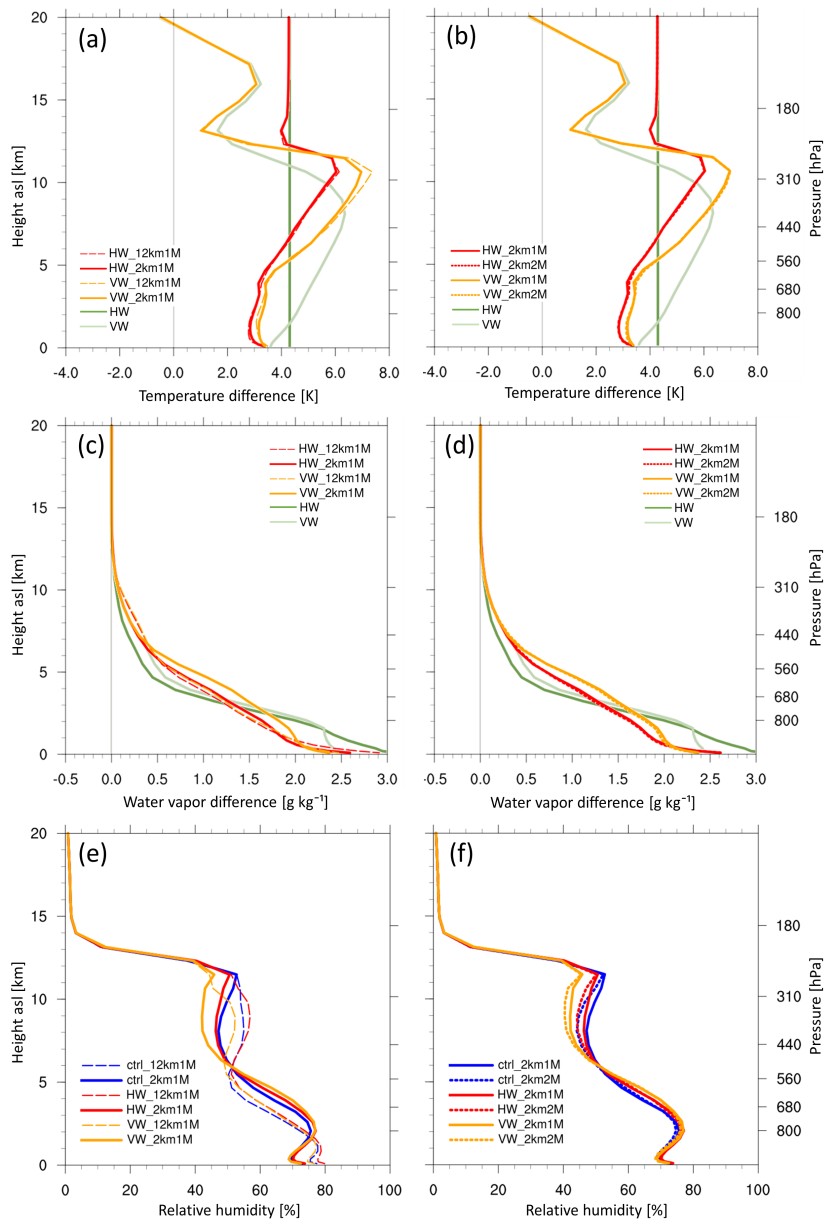

**Figure 2.** Vertical profiles, averaged spatially and over all hours from 3 to 13 June 2007 of (a,b) temperature difference inside the analysis domain of HW (red) and VW (orange) simulations with respect to the control simulations, and at the lateral boundaries of the CPM simulations domain (green, same as Fig. 1b), (c,d) specific humidity difference inside the analysis domain of HW and VW simulations with respect to the control simulations, and at the lateral boundaries of the CPM simulations domain, and (e,f) relative humidity of all nine simulations (control in blue). Dashed lines indicate 12km1M runs (left column), solid lines 2km1M runs (both columns), and dotted lines 2km2M runs (right column). Height is indicated in km on the left side and in hPa on the right side of every graph.





Vertical profiles of specific humidity differences in the analysis domain and at the lateral CPM-boundaries are shown in Figs. 2c and d. These differences are positive, as expected. Similar to temperature, a decrease is found compared to the initial profiles (green) below a certain height and an increase above this height. Here, this height is a little bit lower than with temperature, at 3 km. Also in this case, the difference due to different microphysics schemes is smaller than between the CPM

and the CRM runs.

Vertical profiles of relative humidity are shown in Figs. 2e and f. The strongest differences are again found between the CPM and the CRM runs. Further, VW has lower values than HW between 6 and 12 km, where water vapor content is similar to HW but temperatures are higher.

## 3.2 Precipitation

Before investigating vertical structures of wind and clouds, we document the impact of the surrogate climate change on precipitation, a key component of the hydrological cycle and indicator of convective activity. To give an overview, the spatial distribution of total accumulated precipitation for the 11-day period is shown in Fig. 3, for the observations and the nine simulations. The observations are limited to a region of the European Alps and adjacent areas. Maxima of accumulated observed precipitation are mainly found in the western part of the domain. In the control simulations, precipitation to the north-east of

the Alps is overestimated, in particular in CTRL_12km1M. For all CRM runs, more fine-scale structures are found than for the CPM runs. For HW and VW, areas with values larger than 140 mm increase compared to control, which indicates an increase of heavy precipitation events.

The mean diurnal cycle of precipitation, is shown in Fig. 4. Large differences are found between the 12km1M and 2km1M runs, such as a time shift of three hours (Fig. 4a). This is in line with previous work (e.g. Langhans et al., 2013). The diurnal

cycles of the 2km2M runs are much closer to the 2km1M runs (Fig. 4b) than the 12km1M runs. This similarity in precipitation response between 1M and 2M for the control simulations has already been seen in Keller et al. (2016) and is now also observed in the surrogate warming experiments. The mean precipitation amount of the CPM simulations increases with +14.5 % and +3.4 % for HW and VW, respectively (Table 2). These changes are twice as large as the changes found in Kröner et al. (2016) in 30 years of summer climate. This difference is not surprising as our study focuses on a specific weather situation with a

pronounced diurnal cycle of convection and not on climate means. Of larger interest are the differences between 12 km and 2 km simulations. Indeed, in comparison to 12 km, the response to HW and VW are much smaller in 2 km, both for the 1M and 2M simulations. Both VW surrogate runs even exhibit a decrease in precipitation amount compared to control (Table 2), whereas the corresponding 12 km run shows an increase. In contrast to these simulations, Ban et al. (2015) found a decrease in mean summer precipitation for CPM and CRM simulations. We attribute this reduction to circulation changes with the warmer

climate in their simulations, following the arguments of Kröner et al. (2016).

So far, total accumulated precipitation and the mean diurnal cycle have been investigated. Next, hourly precipitation intensities are analyzed. In Figs. 5a and b, the frequency of grid points with precipitation exceeding a certain threshold is shown. Different dependencies are found for higher and lower thresholds. Intensities with thresholds >45 mm h$^{-1}$ depend mainly on




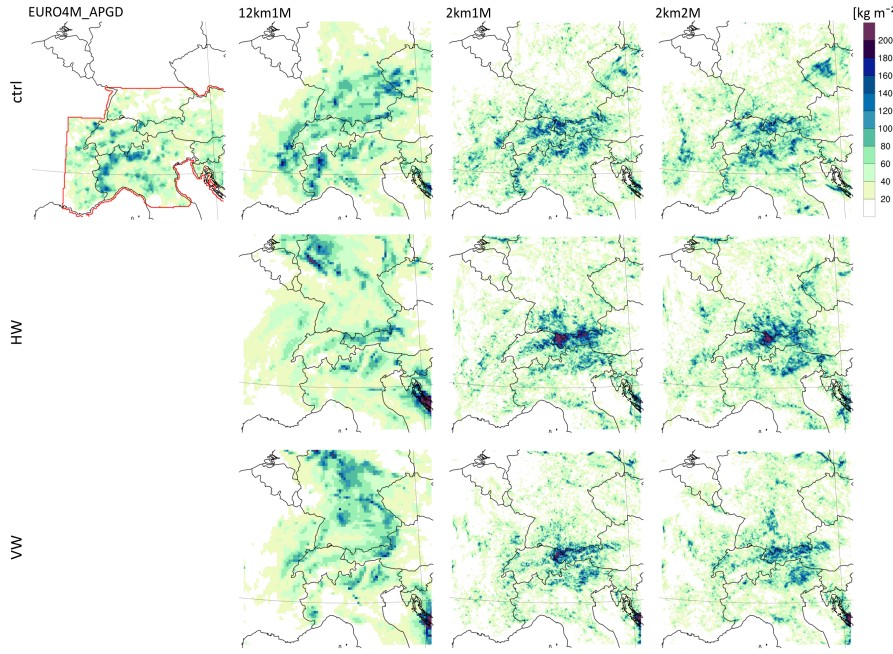

**Figure 3.** Total accumulated precipitation for 3 to 13 June 2007 over the analysis domain for observations EURO4M-APGD and nine simulations. The area of observation is smaller than the model domain and the border is indicated in red. The upper row shows observations and three simulations for the present climate (CTRL), the middle row the three HW simulations, and the lower row the three VW simulations.

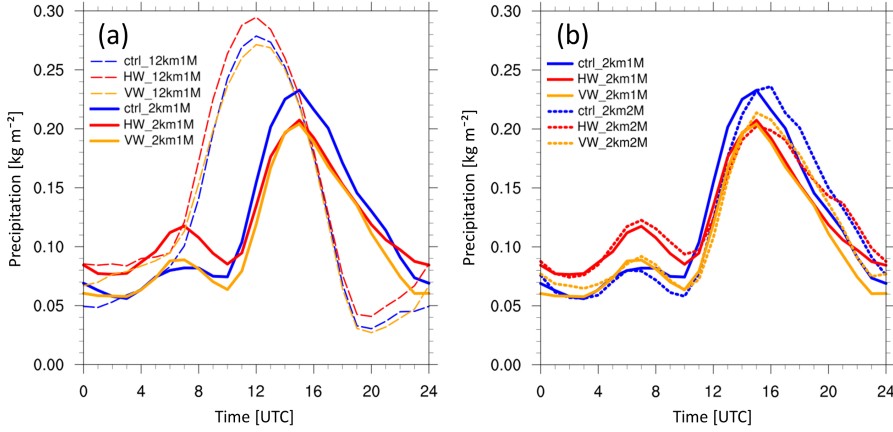

**Figure 4.** Spatially and temporally averaged diurnal cycles of precipitation for 3 to 13 June 2007 (a) for three 12km1M runs (dashed lines) and three 2km1M runs (solid lines), and (b) for the three 2km1M runs (solid lines) and three 2km2M runs (dotted lines). Control runs are indicated in blue, HW runs in red, and VW runs in orange.





**Table 2.** Relative changes in total accumulated precipitation with surrogate warming compared to control in %. The calculations consider spatial means over the analysis domain accumulated for 3 to 13 June 2007.

|     | 12km1M | 2km1M | 2km2M |
| --- | ------ | ----- | ----- |
| HW  | 14.5   | 1.8   | 6.2   |
| VW  | 3.4    | -11.4 | -5.2  |

**Table 3.** The 99.99th percentiles of the relative changes of hourly accumulated precipitation with respect to control, normalized by the averaged 2 m temperature change, for 3 to 13 June 2007. The relative change is called scaling rate ($SR$) and indicated in % K$^{-1}$.

|     | 12km1M | 2km1M | 2km2M |
| --- | ------ | ----- | ----- |
| HW  | 10.0   | 4.0   | 3.9   |
| VW  | 7.7    | 2.4   | 3.5   |

the atmospheric conditions (CTRL, HW, or VW) and the choice of the microphysics scheme (1M or 2M), but intensities with thresholds <25 mm h$^{-1}$ depend mainly on the convection setup and the resolution (12 km or 2 km).

The relative change of hourly accumulated precipitation intensities, normalized by the spatially and temporally averaged 2 m temperature change, is called scaling rate ($SR$) and investigated next (Table 3, Figs. 5c, d, see Sect. 2.2). For the CPM simulations, the increase in extreme precipitation intensities converges with increasing percentiles to values above 7 % K$^{-1}$ (Table 3, Fig. 5c), which is consistent with findings of previous studies (Lenderink and Van Meijgaard, 2008; Kendon et al., 2014). According to the Clausius-Clapeyron relation, values below 6–7 % K$^{-1}$ would be expected (e.g. Allen and Ingram, 2002). Indeed, for the CRM simulations, independent of the microphysics scheme, values below the upper limit of Clausius-Clapeyron are found. The differences between 12km1M and 2km1M are similar to findings of Ban et al. (2015) for their comparison of 10 years simulated summer climate.

### 3.3 Vertical profiles and clouds

In the following, the formation of clouds is investigated. In Figs. 6a and b, vertical profiles of grid-scale vertical velocity are shown, which are split into averages over the negative and positive components. Mean upward motion is slightly larger than mean downward motion, but the values are comparable. In Fig. 6a, large differences are found between CPM and CRM simulations. This is because for CPM, a part of the vertical transport is calculated inside the convection scheme and not represented by the grid-scale vertical wind component used for this analysis. Apart from this difference, the values for all CPM simulations and all CRM simulations are comparable. Below 10 km, the lowest values are found for VW. Above 10 km, higher values are found for HW and VW than for control with the highest values for VW. In Fig. 6b, similar results are found for all 2 km simulations. In more detail, grid-scale vertical velocity is larger for 2km2M than 2km1M above 1500 m height. This result differs from the results of Fig. 5b where more heavy precipitation events for 2km1M than 2km2M were found. But in contrast



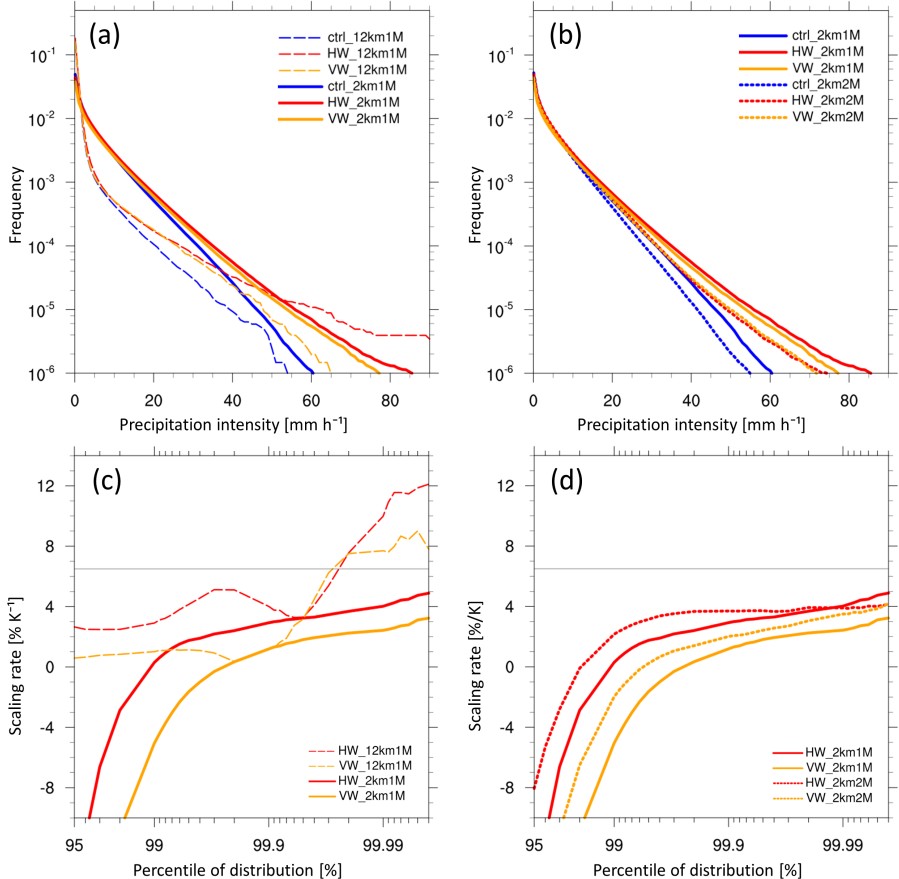

**Figure 5.** (a,b) Frequency of hourly accumulated precipitation intensity for 3 to 13 June 2007 for (a) three 12km1M runs (dashed lines) and three 2km1M runs (solid lines), and (b) the three 2km1M runs (solid lines) and three 2km2M runs (dotted lines). (c,d) $SR$, as a function of percentile, in % K$^{-1}$. Control runs are indicated in blue, HW runs in red, and VW runs in orange. The gray lines in (c) and (d) indicate the expected upper limit (6–7 % K$^{-1}$) according to the Clausius-Clapeyron relation.

to Fig. 5b, only mean values are shown in Fig. 6b. In comparison to mean precipitation amounts (Table 2), the increased values for 2km2M are not surprising.

Figs. 6c and d show vertical profiles of grid-scale cloud water ($q_c$) and cloud ice ($q_i$) content. For 12km1M, higher values are found for $q_i$ than for $q_c$. For 2km1M and 2km2M, the situation is opposite. The largest difference between 2km1M and 2km2M is found with the vertical distribution of $q_i$. Due to ice sedimentation, $q_i$ is also found at lower altitudes for 2km2M. For all simulations, the amounts of $q_i$ are similar for control and HW but reduced for VW. The highest values of $q_c$ are found for HW with 1M and for CTRL with 2M. Further, HW and VW have higher values above the peaks than control. Note that subgrid clouds are not considered in the calculation of $q_c$ and $q_i$.





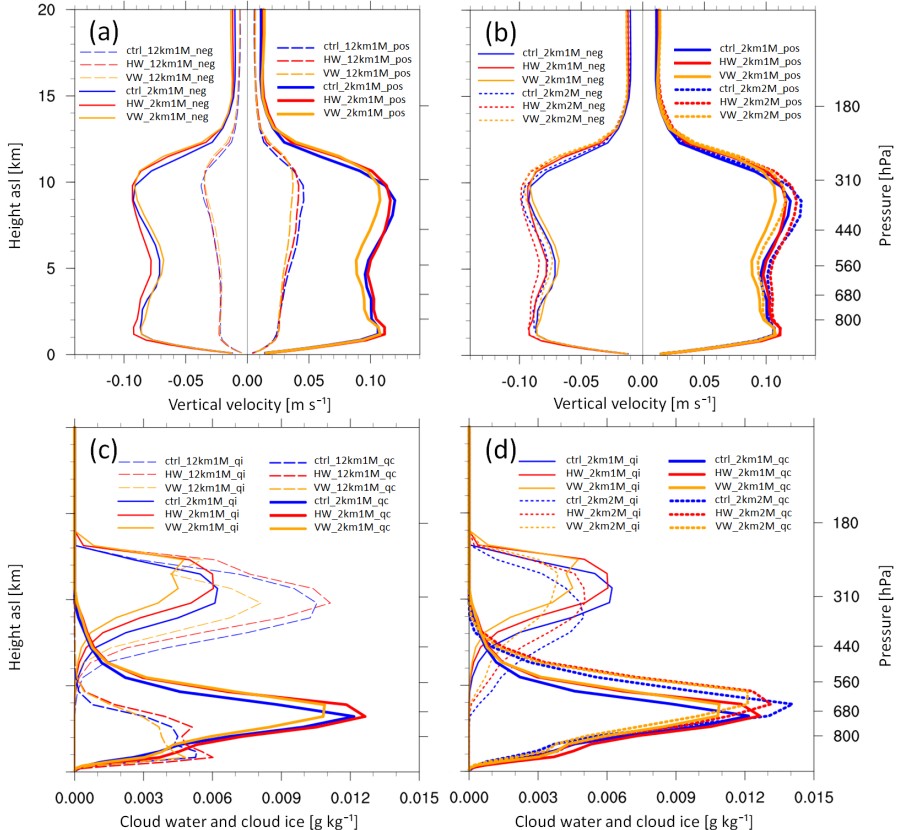

**Figure 6.** Vertical profiles of (a,b) grid-scale vertical velocity and (c,d) cloud ice content ($q_i$) and cloud water content ($q_c$) from nine simulations, averaged horizontally over the full analysis domain and temporally over all hours from 3 to 13 June 2007. Control runs are indicated in blue, HW runs in red, VW runs in orange, 12km1M runs with dashed lines (left column), 2km1M runs with solid lines (both columns), and 2km2M runs with dotted lines (right column). Vertical velocity is divided into averages of negative and positive values. Height is indicated in km on the left side and in hPa on the right side of every graph.

The influence of the surrogate climate change on clouds (including subgrid clouds) is investigated with two-dimensional histograms of cloud optical thickness (COT) and cloud top pressure (CTP) at 13 UTC (cf. Keller et al., 2016). These histograms define several cloud types. After Rossow and Schiffer (1999), high, middle, and low clouds are distinguished at 440 and 680 hPa, and thin, middle, and thick clouds at 3.6 and 23 COT. In Fig. 7, a positive bias in high clouds and a negative bias in mid-level clouds are found for CTRL_12km1M and CTRL_2km1M compared to the observations. For CTRL_2km2M, the bias in high cloud occurence is small and differences are mainly found regarding the thicknesses of these clouds. For the mid-level clouds, the bias of CTRL_2km2M is similar to the bias of CTRL_2km1M. The negative bias in mid-level clouds coincides with the low values of $q_c$ and $q_i$ around 6 km height in Figs. 6c and d. Note that the histogram, calculated from observational data, shows some differences with respect to a previously published version (Keller et al., 2016). This is partly





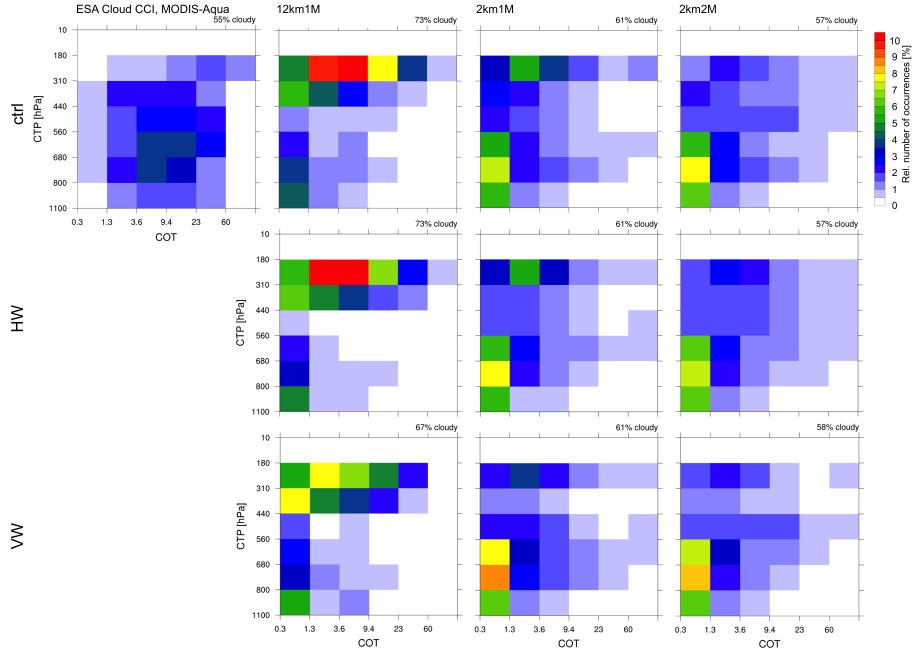

**Figure 7.** Histograms of cloud frequency as a function of cloud optical thickness (COT) and cloud top pressure (CTP) arranged as for Fig. 3 but as an average at 13 UTC (for the models) over the period of 3 to 13 June 2007. For observation, the average is over local time of the Aqua satellite (approx. 01:30 pm). Fractional cloud cover (defined by COT > 0.3) is indicated in the right upper corner of all panels.

due to the use of raw data from a different satellite sensor but mainly due to a revised algorithm to produce the dataset (version 2.0 versus version 1.0). Between control and HW, only small differences are found, but a substantial reduction in high clouds is found for VW. Therefore, the thermodynamic effect (CTRL to HW) is small in this case, but the lapse rate effect (HW to VW) is large. The strong similarity between CTRL and HW is a surprising result, since higher amounts and intensities in

precipitation were found for HW than for control. The reduction of high clouds in VW cannot be explained completely with the reduction of vertical velocity below 310 hPa (Figs. 6a, b) because this reduction is very small. Moreover higher vertical velocity is found above this height compared to control. But the reduced relative humidity, found for VW compared to control and HW at these heights (Figs. 2e, f) due to higher temperatures (Figs. 2a, b) and similar specific humidity (Figs. 2c, d), may explain the reduced high cloud frequency. We assume that the similar amounts of specific humidity of HW and VW close to

the ground, where most water vapor is found, lead to similar absolute water content in convective updrafts of both cases but in VW convective condensation is reduced due to the higher temperatures at higher levels.

The diurnal cycles of cloud cover and ground temperature impact outgoing longwave radiation (OLR) (Figs. 8a, b). Negative mean biases for CTRL_12km1M and CTRL_2km1M, a slightly positive mean bias for CTRL_2km2M, and a delay in the diurnal cycles of all control simulations are found compared to the observations. The timing of OLR stays the same with the

15 surrogate runs, which indicates a similar timing in the diurnal cycle of clouds. The warmer cloud temperatures due to the



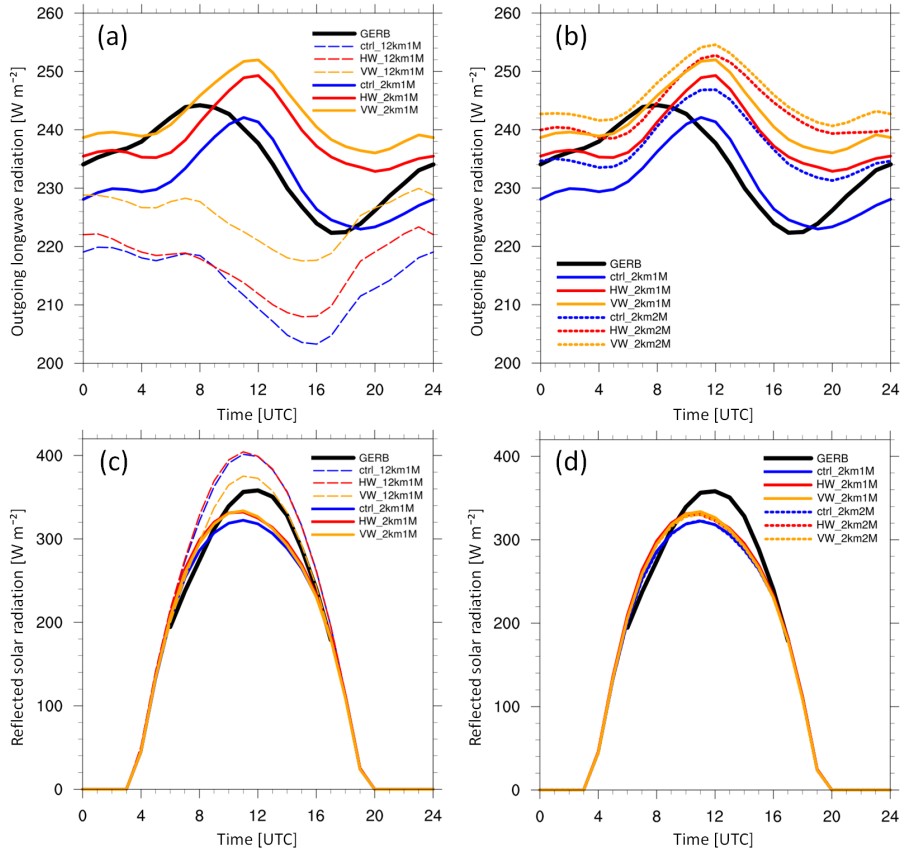

**Figure 8.** Spatially and temporally averaged diurnal cycles of (a,b) outgoing longwave radiation (OLR) and (c,d) reflected solar radiation (RSR) for observations from GERB and nine simulations for 3 to 13 June 2007. Observations are indicated in black, control runs in blue, HW runs in red, VW runs in orange, 12km1M runs with dashed lines (left column), 2km1M runs and observations with solid lines (both columns), and 2km2M runs with dotted lines (right column).

increased surrounding air temperatures and the warmer ground temperatures of HW and VW, and further the reduction of high cloud cover for VW lead to larger OLR mean values than for control. Reflected solar radiation (RSR) is mainly impacted by changes in cloud cover (Figs. 8c, d). It is underestimated in CTRL_2km1M and CTRL_2km2M but overestimated in CTRL_12km1M. In addition, all control simulations show a too early peak. With the surrogate simulations, the diurnal cycles

5  do not change in timing but in amplitude, particularly for VW_12km1M due to the reduced cloud cover.

For the energy budget at the top of the atmosphere (ToA), consisting of the sum of OLR and RSR, rather small changes are found for HW_12km1M and VW_12km1M compared to CTRL_12km1M with 4.0 and –0.8 W m$^{-2}$, respectively. For HW_2km1M and VW_2km1M compared to CTRL_2km1M, much larger changes are found with 11.8 and 14.0 W m$^{-2}$, respectively. For HW_2km2M and VW_2km2M compared to CTRL_2km2M, the situation is similar to 2km1M with 9.9 and

10  10.8 W m$^{-2}$, respectively. These values of the ToA energy budget are summarized in Table 4. Therefore, the CPM runs sug-





**Table 4.** Changes in the ToA energy budget (OLR + RSR) with surrogate warming compared to control in W m$^{-2}$.

|     | 12km1M | 2km1M | 2km2M |
|-----|-------:|------:|------:|
| HW  | 4.0    | 11.8  | 9.9   |
| VW  | -0.8   | 14.0  | 10.8  |

gest that the surrogate warming has no or only a small impact on the ToA energy budget during this period. In contrast, the CRM runs suggest a much larger increase in outgoing energy fluxes, and therefore a cooling of the heated atmosphere. These differences are crucial, as they will influence the results of long-term climate simulations.

## 4 Conclusions

5 The impact of surrogate climate change on precipitation and clouds has been investigated for an 11-day period with a pronounced diurnal cycle of convection. Two different warming scenarios are considered: a homogenous warming (HW) and a vertically-dependent warming (VW). The surrogate approach has been successfully applied to convection-resolving model (CRM) simulations. The CRM simulations at 2.2 km resolution are complemented by convection-parameterizing model (CPM) simulations at 12 km resolution. These simulations use a one-moment microphysics scheme (1M), while the CRM simulations are also available with a two-moment microphysics scheme (2M). The 2M is used due to the positive impact of its ice sedimentation on the high cloud cover bias, which was found in Keller et al. (2016). To our knowledge, this is the first application of the surrogate approach for Alpine summer climate using CRM simulations.

For the CRM simulations, an increase in hourly heavy precipitation events is found for both surrogate warming experiments (HW and VW) compared to control, independent of the microphysics scheme. These increases are consistent with the Clausius-Clapeyron relation. In contrast, the CPM simulations exhibit a stronger increase in heavy precipitation events. This difference between CPM and CRM simulations has previously been noted in Ban et al. (2015).

The vertical structure of the warming, represented by HW and VW, has a significant impact on the clouds of the diurnal cycle of convection. For both microphysics schemes, the clouds of HW experience virtually no change compared to control, apart from changes in their temperature. On the other hand, the amount of high clouds of VW is reduced, indicating a strong influence of the lapse rate on cloud cover. This change in cloud cover is consistent with the role of the lapse rate for convection. Despite these differences in cloud type frequencies between HW and VW, all four surrogate simulations with 2 km resolution suggest a cooling effect in the energy budget at top of the atmosphere with 9.9 to 14.0 W m$^{-2}$ compared to control. The corresponding increases in the energy budget of the 12 km simulations are much smaller, and amount to merely between -0.8 and 4 W m$^{-2}$.

The results of the CPM simulations largely coincide with the CRM results. However, the significant differences in the response to the HW and VW forcing between the two setups underline the importance to complement CPM simulations with CRM simulations.





Finally, there are large differences between CPM and CRM in terms of timing and amplitude of the diurnal cycle of precipitation, and regarding the occurrence of low precipitation intensities. Previous studies have shown that the CRM simulations generally validate better against observations. It is worth noting, however, that these differences are larger than the sensitivity of CTRL with respect to HW and VW.

5 ## 5    Data availability

The Cloud_cci data is publicly accessible at www.esa-cloud-cci.org. The EURO4M-APGD data can be ordered from MeteoSwiss at dx.doi.org/10.18751/Climate/Griddata/APGD/1.0. The GERB data can be accessed after a registration is accepted at http://gerb.oma.be/ →"Data Access (ROLSS)" →"Register to the ROLSS mailing list".

*Competing interests.*    The authors declare that they have no conflict of interest.

10 *Acknowledgements.*    We would like to thank Nikolina Ban and Axel Seifert for helpful comments. We also thank the GERB project team for providing access to their data. The numerical simulations have been performed at the Swiss National Supercomputing Centre (CSCS). This work was financially supported by ETH Research Grant CH2-01 11-1 and co-funded by the European Space Agency through the Cloud_cci project (contract No.: 4000109870/13/I-NB).



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
