# Peer review of "The sensitivity of Alpine summer convection to surrogate climate change: An intercomparison between convecton-parameterizing and convection-resolving models"

_Atmospheric Chemistry and Physics, 2017_

## Referee Comment (RC1) · Anonymous Referee #1 · 16 Aug 2017

General

This is a very nice process study concerning summer convection in the Alps under climate change. The authors use the surrogate climate change approach and compare intermediate resolution (convection-parameterizing) and high-resolution (convection-resolving) simulations for an 11-day period in summer 2017. The paper is well written and clear, the employed methodology well established and sound, the presentation is

straight forward and the presented material (figures, tables) appropriate and (in most cases) clear. The only drawback of the paper is that it essentially produces already known results (even in the abstract the authors write, e.g., '...exhibit substantial differences, which are well known from the literature') - but with a not yet used approach. I therefore have only one 'major comment', which invites the authors to better work out what actually is the benefit from this study (it should be noted that also replicating a result with another approach has a value in itself). Apart from this I have a number of minor comments – and overall I think the paper can be published subject to 'minor revisions'.

Major comment

In the introduction, the authors give a very nice (and quite comprehensive) overview on the state the art in regional climate modeling with respect to summertime convection in the alps. Essentially, CPM and CRM resolutions had been compared (e.g., Ban et al. 2015), the surrogate climate approach (SCA) has been used before (Kroener et al. 2017), the two microphysics schemes had been compared (Keller et al. 2016). What is new in the present study according to the authors (p16, l.11) is the use of the SCA for CRM. So, one would expect to learn a bit more on the advantages of the SCA in the first place (what is it that we can learn from it that we cannot obtain from the decade-long CRM simulations?) – and why? Second, one would expect the approach to be put into perspective: open questions according to the introduction are i) the limitation of precipitation extremes due to the Clausius-Clapeyron equation (p2, l.25), ii) the role of stratification [changes] for the precipitation in CRM (p3, l.2) and iii) the impact of the microphysics scheme on the cloud top [not so much the precipitation], (p3, l.11). So, which of these topics can be better addressed with the SCA than with decade-long simulations (and why)? What can we learn from a 11-days process study that cannot be obtained from the full decade-long simulations? At the end of the introduction, the authors then formulate three questions they want to address (p3, l.14). These questions, however, do not correspond (one-to-one) to those open questions – and to

some degree do also not reflect what the authors summarize in their conclusions. I think therefore, the authors could make a much stronger case for their simulations if they would thoroughly work out what the potential of their simulations (approach) is (more than 'it has not been done') and by discussing their results in the light of earlier findings (and whether a disagreement could be resolved or explained) and their own questions in the beginning.

Minor comments

P2, l.27 I think the authors should expand a little on the advantages of the surrogate warming (and also maybe on the disadvantages). While in 1996 this approach was certainly mainly advantageous with respect to computing time, this has changed a little in recent times.

P6, l.25 yielded a bimodal. . ..

P7, l.14 Indeed, the 11-day period is quite limiting. Maybe the authors can expand a little on what processes they want to explore in more detail than possible with the decade-long CRM simulations. And especially, how this is related to the potential of the SCA.

P7, l.28 we assume. . .: of course, because this is a paper about convection - but couldn't this hypothesis be checked more thoroughly (using all the available data)?

P9, l.22 here we also have observations - so it would be interesting to see the mean diurnal cycle of the observations, too (this could be realized by also in the model only considering the 'observed part' of the domain - assuming [but the authors can of course judge, based on the results], that the mean daily cycles in the model runs will not largely change if only a subdomain is used).

P9, l.26 first of all: the responses to HW and VW are... . More important: I am not sure what the authors want to point out here. Under the 'response to HW and VW' I would intuitively understand the difference between the blue curves on the one hand, and the

red/orange lines on the other hand. If we take the peak, the dashed lines in Fig. 4a are rather closer to each other (i.e., the red/orange closer to the blue) than the full lines. So, is it something else that the authors want to point out? Apparently, when reading on, the authors refer to mean, temporally averaged precipitation (as given in Tab. 2). In a paper that deals with convection, however, I would find it extremely noteworthy, that HW is only larger during the night (and the microphysics scheme doesn't change this). So, what I really find striking is the large difference between HW and VW during the night - irrespective of all other differences. Can the authors comment on this?

P9, l.28 'these' are the present simulations, right?

P9, l.29 we attribute 'this' reduction': which now? the CPM, or the CRM? those of Ban et al. or the present? Please specify (Note that if the present simulations were referred to, an appropriate comment would have been that this could be more than a hypothesis - because the authors have all the simulation data so they could identify the circulation changes over the 11 days...).

P9, l.30 interestingly, in my paper collection this is Kroener et al 2017.Âă (Clim Dyn (2017) 48:3425–3440, DOI 10.1007/s00382-016-3276-3).

P11, l.2 what is the 'convection setup'?

P11, l.9 see above: most findings are 'similar to those of Ban et al'. → by elaborating a little bit more on why to use this surrogate climate approach (even if for only 11 days) this would help to better motivate the present study.

P13, l.4 couldn't those 'discrimination lines be shown in the figure?Âă

Fig. 7 the smallest letters and numbers are definitely too small.

P14, l.6 'below 310 hPa' seems to be misleading (I intuitively first checked p<310 hPa...). Better would be 'for heights below 310 hPa'.

P14, l.15 . . .indicates a similar timing. . .: Still it is interesting to note that precipitation

peaks around 1500 UTC (Fig. 4) for the CRM simulations and 1200 UTC for the CPM simulations. Combining earlier statements in the paper (p2, l.14, and p9, l.19, i.e. the reference to the Langhans et al. paper where it had been shown that the precipitation maximum is better simulated in higher resolution, and therefore should occur closer to 1500 UTC), this would imply that - at least for the chosen 11-day period - precipitation peaks almost [1-2 hr difference] when cloudiness has its minimum (Fig. 8). This is why (among others) I have suggested to add observed precipitation to Fig. 4. Maybe it can also trigger some additional analysis concerning which type of clouds actually contribute to OLR and to what degree OLR is determined by clouds.

Fig. 8, caption: '(a, b) outgoing longwave radiation at the top of the atmosphere (obs) and the model domain (mod), respectively (I presume).

P15, l. 4 'All the control simulations show a too early peak': isn't this a contradiction to the finding in OLR? If the peak is too early, then the cloud cover peak is too early (but the OLR peak is too late). Can the authors comment on that?

P16, l. 24-26 The results of the CPM. . ..: I think these three lines contradict themselves. First, they can only refer to the VW/HW discussion, and second, if the two largely coincide why then do we have significant differences?

---

## Referee Comment (RC2) · Anonymous Referee #2 · 18 Sep 2017

Major comments: This is a potentially interesting study that deserves publication if the three major changes listed below are made: 1. The number of lines on each plot is too high to properly absorb the message you are trying to communicate. For instance, figure 6 has 12 lines on four plots for 48 lines. While I appreciate that you are trying to make a point that many of the lines are on top of each other, I have a hard time distinguishing the lines from each other and also the difference between one line or another and the key from one line to another. One suggestion is to delete

the comparison to the one moment and two moment microphysics schemes from this paper. There is very little sensitivity and most of your points were made in the previous paper. I would focus on the difference between the 2.2 km and 12 km simulations and the current and surrogate climate.

2. The paper treats all the results with nearly equal weight. I find it to more of a travelogue than a research paper. I think you have a message you want to convey to the reader and I would focus on that message from both the figures you show and the discussion in the text. As I mention above, I was most interested in the difference between the convective permitting and the convection parameterization simulations for both current and future climate. This message is lost in the travelogue style of presentation.

3. I would like to see more emphasis on the physical reason for the results. For instance, why is the diurnal timing for convection changed going from convective permitting to convective parameterization simulations in figure 4? Why does the CPM simulation have less precipitation? Is the amount it estimated close to observations? Otherwise these are just model results and I haven't really learned anything other than there is a difference between the runs or not. There are only 11 days of simulation, so a focus on the physics rather than the climatology seems warranted and appropriate.

Minor comments: Page 3, line 1. Please state the height in the atmosphere for which the north-south temperature gradient is impacted. Page 6. Line 29. Delete "steps at". Page 6. I would like to see an image of the analysis domain in this paper. Page 7, line 8. I would have liked to have seen a sequence of synoptic maps characterizing the 10 day period (if nearly constant, a composite map). Page 9. Line 22."with" should be replaced by "by". Page 10. I would like to see difference plot for figure 3. It is very difficult to see what the differences in various runs actually are and what magnitude otherwise. Page 9, line 29. Can you be more clear about the expected cirdulation changes? Page 11, line 9. Need to state what the differences are between the 12 km and 2 km runs and how they agree with Ban et al. (2015). Page 11, line 14. Need to state what

the values of vertical velocity you are talking about. Page 14. Lines 3-14. This is an interesting discussion of the causes for the differences in the VW and HW simulations. I would encourage a more detailed analysis as the discussion speculates more than determines what the real cause is. It might be useful to examine the evolution of clouds in detail for one or two days for both VW and HW to determine the cause. You only have 11 days, so an average does not necessarily give you a robust result. Page 16. Lines 15 and 16. Why is there an increase in heavy precipitation events for the CPM runs compared to the CRM runs? Do the CPM runs compare well to the CRM runs for non-heavy events? Page 16, lines 24-26. I think this is the most interesting result of the paper and should be explored deeper. First of all, how does the VW change in vertical distribution physically effect the clouds? You only have 11 days of simulation, so you should be able to note some common evolution and physical changes. Second, why is the response of the CRM different than the CPM for HW and VW? This is a very interesting result that deserves more investigation.

Final comment: The conclusion section is much too short. There should be much deeper discussion of the results here that can help the reader understand the detailed simulation results presented in the previous section. What do you want the reader to take away from this study? I am current not sure, and that is a problem.

---

## Author Comment (AC1) · 19 Dec 2017

The comment was uploaded in the form of a supplement:
https://www.atmos-chem-phys-discuss.net/acp-2017-504/acp-2017-504-AC1-supplement.pdf

2017.

---

## Author Response (AR1)

**Letter to the Editor:**

Before addressing the two reviews, we would like to point out that because of an oversight during cleaning older simulation data, we removed a part of the 3D model output, which is needed to reproduce Figs. 2 and 6 of this article. We therefore repeated the simulations discussed in this article using a new hardware configuration following an upgrade of our supercomputer center. The general conclusions and all main results of the article **are not affected** by this. Due to the chaotic nature of the studied phenomena there are a number of differences, including some substantial changes in precipitation amounts. For consistency, all numbers and figures have been updated for the revised version of the paper.

**Response to Referee #1 (acp-2017-504)**

We thank the referee for all the valuable comments that have improved the manuscript. Following his/her suggestions, we have streamlined our paper and focused on specific results, highlighting the benefits of this study. Please see below our point-by-point replies to the specific comments, with the referee's comments in black and our replies in blue.

**Major comment:** In the introduction, the authors give a very nice (and quite comprehensive) overview on the state the art in regional climate modeling with respect to summertime convection in the Alps. Essentially, CPM and CRM resolutions had been compared (e.g., Ban et al. 2015), the surrogate climate approach (SCA) has been used before (Kroener et al. 2017), the two microphysics schemes had been compared (Keller et al. 2016). What is new in the present study according to the authors (p16, l.11) is the use of the SCA for CRM. So, one would expect to learn a bit more on the advantages of the SCA in the first place (what is it that we can learn from it that we cannot obtain from the decade-long CRM simulations?) – and why? Second, one would expect the approach to be put into perspective: open questions according to the introduction are i) the limitation of precipitation extremes due to the Clausius-Clapeyron equation (p2, l.25), ii) the role of stratification [changes] for the precipitation in CRM (p3, l.2) and iii) the impact of the microphysics scheme on the cloud top [not so much the precipitation], (p3, l.11). So, which of these topics can be better addressed with the SCA than with decade-long simulations (and why)? What can we learn from a 11-days process study that cannot be obtained from the full decade-long simulations? At the end of the introduction, the authors then formulate three questions they want to address (p3, l.14). These questions, however, do not correspond (one-to-one) to those open questions – and to some degree do also not reflect what the authors summarize in their conclusions. I think therefore, the authors could make a much stronger case for their simulations if they would thoroughly work out what the potential of their simulations (approach) is (more than 'it has not been done') and by discussing their results in the light of earlier findings (and whether a disagreement could be resolved or explained) and their own questions in the beginning.

We agree with the referees that the paper was lacking a strong focus. Following the suggestions of the second referee we removed the comparisons of the one and two-moment microphysics scheme and further the analysis concerning the Clausius-Clapeyron scaling. With those major changes we could streamline the paper and put more emphasize on the selected findings. Overall this has led to a shorter paper, substantial decrease in figure count, and has contributed to a clearer structure and focus. We want to answer the following questions: How will the diurnal cycle of convection and the associated precipitation and clouds change in a warmer climate? How large is the impact of different temperature change profiles (HW versus VW)? How do the simulated changes depend on the modeling framework, in particular on the horizontal resolution (CPM versus CRM simulations)? To this end we focus on the comparisons between 2.2 and 12km resolution and the two climate change

profiles. Accordingly we focused our introduction on these points and extended our conclusion section to reflect these changes.

Removing the two-moment microphysics has dropped the following panels: Fig. 2B,d,f; Right column of Fig. 3; Fig.4b; Fig.5b; Fig6b,d; Right column of Fig. 7; Fig. 8b,d; The microphysics column in table 1 and the right column of table 2 as well as all references to those figures and their discussion.

Removing the analysis of the Clausius-Clapeyron scaling has dropped the following elements: Fig. 5c,d; the method section 2.2 and table3 as well as all references to those figures and their discussion.

Addressing the first comment from the referee cited above, we extended the introduction to better explain the benefits of the SCA for this study:

See Page 2, Lines 24-28; Page 3, Lines 6-11

Addressing the second comment we focused the paper on the questions formulated at the end of the introduction, and to prevent confusion deleted the Clausius-Clapeyron paragraph as well as the discussion of two-moment microphysics from the introduction. Further we also extended the conclusion section to better reflect the questions we formulated in the introduction.

See Page 4, Lines 1-4, See conclusion section on Pages 13-14

**Minor comments**

Page 2, Line 27: I think the authors should expand a little on the advantages of the surrogate warming (and also maybe on the disadvantages). While in 1996 this approach was certainly mainly advantageous with respect to computing time, this has changed a little in recent times.

See reply to the first point of the major comment above. Further we included a paragraph in the discussion section addressing the disadvantage of the surrogate method. Page 14 Lines 22-24.

Page 6, Line 25 yielded a bimodal. . ..

Text corrected.

P7, l.14 Indeed, the 11-day period is quite limiting. Maybe the authors can expand a little on what processes they want to explore in more detail than possible with the decade-long CRM simulations. And especially, how this is related to the potential of the SCA.

We complemented the text by explaining the investigated process and the advantage of the surrogate approach. Page 3 Lines 6-11 and Page 6, Line 32 and Page 7, Lines 1-2

P7, l.28 we assume...: of course, because this is a paper about convection - but couldn't this hypothesis be checked more thoroughly (using all the available data)?

We have reformulated this sentence, in the revised version (Page 7, Liens 13-14) we now state that the vertical redistribution of temperature "must largely be caused by convection and boundary layer processes". We consider it unlikely that other processes (such as radiation) do also contribute. The full analysis would actually be quite cumbersome, as the parameterized and advective tendencies would need to be used (which mostly are not available with the current model output).

P9, l.22 here we also have observations - so it would be interesting to see the mean diurnal cycle of the observations, too (this could be realized by also in the model only considering the 'observed part' of the domain - assuming [but the authors can of course judge, based on the results], that the mean daily cycles in the model runs will not largely change if only a subdomain is used).

The observations in Figure 3 are daily means. For a comparison of the diurnal cycle of precipitation, hourly data from some subdomains (e.g. Switzerland) could be used. However, since we show the comparison of the diurnal cycle with Swiss observations in Keller et al. 2016, we decided against including a detailed validation of the diurnal cycle of precipitation observations in the current paper. To make the reader aware of the validation in the previous paper, we make a specific comment (see Page 9, Lines 1-2).

P9, l.26 first of all: the responses to HW and VW are... . More important: I am not sure what the authors want to point out here. Under the 'response to HW and VW' I would intuitively understand the difference between the blue curves on the one hand, and the red/orange lines on the other hand. If we take the peak, the dashed lines in Fig. 4a are rather closer to each other (i.e., the red/orange closer to the blue) than the full lines. So, is it something else that the authors want to point out? Apparently, when reading on, the authors refer to mean, temporally averaged precipitation (as given in Tab. 2). In a paper that deals with convection, however, I would find it extremely noteworthy, that HW is only larger during the night (and the microphysics scheme doesn't change this). So, what I really find striking is the large difference between HW and VW during the night - irrespective of all other differences. Can the authors comment on this?

We reformulated the paragraph so that it can be understood more easily. Page 9 Lines 5-13. We were also surprised by the HW effect on precipitation during night. The destabilizing effect of HW seems to be more pronounced during night time. Currently we do not have a convincing explanation for this behaviour.

P9, l.28 'these' are the present simulations, right?

Yes. We replaced 'these' with 'our' for a better understanding.

P9, l.29 we attribute 'this' reduction': which now? the CPM, or the CRM? those of Ban et al. or the present? Please specify (Note that if the present simulations were referred to, an appropriate comment would have been that this could be more than a hypothesis - because the authors have all the simulation data so they could identify the circulation changes over the 11 days...).

We reformulated the whole paragraph. Page 9 Lines 5-13

P9, l.30 interestingly, in my paper collection this is Kroener et al 2017. (Clim Dyn (2017) 48:3425–3440, DOI 10.1007/s00382-016-3276-3).

Reference updated.

P11, l.2 what is the ‚convection setup'?

"convection setup" was referring to the horizontal resolution. We reformulated the paragraph.

P11, l.9 see above: most findings are 'similar to those of Ban et al'. → by elaborating a little bit more on why to use this surrogate climate approach (even if for only 11 days) this would help to better motivate the present study.

This paragraph was removed. We extended the introduction and method section to work out the benefits of the SCA approach. See answer to major comment.

P13, I.4 couldn't those 'discrimination lines be shown in the figure?

We also thought about that but (1) adding another two lines to figure 7 makes the plot even more busy, and (2) the axes in figure 7 are chosen such that the 440 and 680 hPa level can easily be seen.

Fig. 7 the smallest letters and numbers are definitely too small.

We increased the figure size

P14, I.6 'below 310 hPa' seems to be misleading (I intuitively first checked p<310 hPa...). Better would be 'for heights below 310 hPa'.

Suggestion followed.

P14, I.15 . . .indicates a similar timing. . .: Still it is interesting to note that precipitation peaks around 1500 UTC (Fig. 4) for the CRM simulations and 1200 UTC for the CPM simulations. Combining earlier statements in the paper (p2, I.14, and p9, I.19, i.e. the reference to the Langhans et al. paper where it had been shown that the precipitation maximum is better simulated in higher resolution, and therefore should occur closer to 1500 UTC), this would imply that - at least for the chosen 11-day period - precipitation peaks almost [1-2 hr difference] when cloudiness has its minimum (Fig. 8). This is why (among others) I have suggested to add observed precipitation to Fig. 4. Maybe it can also trigger some additional analysis concerning which type of clouds actually contribute to OLR and to what degree OLR is determined by clouds.

During our period of interest, the OLR minima at around 17~UTC, indicates a maximal extension of high, cold clouds and corresponds therefore well with the maximum of precipitation, which is expected a little bit earlier than the cloud maxima. We would attribute the OLR maxima (Fig. 7a) around noon for the CRM simulations, which is still 2 hours before peak precipitation, to the rising atmospheric and ground temperatures during the day. After 12 UTC, this effect is overcompensated by the increasing cloud cover, which decreases OLR. It is important to notice that the 12km simulations are not able to reproduce these maxima over day between 8-12 UTC but have maxima at midnight (Fig 7a). For an in-depth analyses of the cloud cover changes, see Keller et al. 2016.

We reformulated the whole paragraph: Page 12, Lines 16-19 and Page 13, Lines 1-9

Fig. 8, caption: '(a, b) outgoing longwave radiation at the top of the atmosphere (obs) and the model domain (mod), respectively (I presume).

Caption revised

P15, I. 4 'All the control simulations show a too early peak': isn't this a contradiction to the finding in OLR? If the peak is too early, then the cloud cover peak is too early (but the OLR peak is too late). Can the authors comment on that?

Indeed, the formulation was misleading. An increase of high clouds leads to lower values in OLR and higher values in RSR. The too early peak in RSR in the morning corresponds to the OLR minimum in the morning. Further, OLR is sensitive to different cloud heights, while RSR is less sensitive to that. We complemented the text with more details. Page 13 Lines 6-9

P16, l. 24-26 The results of the CPM. . ..: I think these three lines contradict themselves. First, they can only refer to the VW/HW discussion, and second, if the two largely coincide why then do we have significant differences?

We agree the text was not clearly formulated we changed large parts of the conclusions.

**Response to Referee #2 (acp-2017-504)**

We thank the referee for all the constructive comments that have improved the manuscript. We followed the suggestions of the referee and reduced the amount of presented results giving more focus on specific ones. Please see below our point-by-point replies to the comments, with the referee's comments in black and our replies in blue.

**Major comments**

This is a potentially interesting study that deserves publication if the three major changes listed below are made:

We agree with the referees that the paper was lacking a strong focus. Following the suggestions of the second referee we removed the comparisons of the one and two-moment microphysics scheme and further the analysis concerning the Clausius-Clapeyron scaling. With those major changes we could streamline the paper and put more emphasize on the selected findings. Overall this has led to a shorter paper, substantial decrease in figure count, and has contributed to a clearer structure and focus. We want to answer the following questions: How will the diurnal cycle of convection and the associated precipitation and clouds change in a warmer climate? How large is the impact of different temperature change profiles (HW versus VW)? How do the simulated changes depend on the modeling framework, in particular on the horizontal resolution (CPM versus CRM simulations)? To this end we focus on the comparisons between 2.2 and 12km resolution and the two climate change profiles. Accordingly we focused our introduction on these points and extended our conclusion section to reflect these changes.

Removing the two-moment microphysics has dropped the following panels: Fig. 2B,d,f; Right column of Fig. 3; Fig.4b; Fig.5b; Fig6b,d; Right column of Fig. 7; Fig. 8b,d; The microphysics column in table 1 and the right column of table 2 as well as all references to those figures and their discussion.

Removing the analysis of the Clausius-Clapeyron scaling has dropped the following elements: Fig. 5c,d; the method section 2.2 and table3 as well as all references to those figures and their discussion.

1. The number of lines on each plot is too high to properly absorb the message you are trying to communicate. For instance, figure 6 has 12 lines on four plots for 48 lines. While I appreciate that you are trying to make a point that many of the lines are on top of each other, I have a hard time distinguishing the lines from each other and also the difference between one line or another and the key from one line to another. One suggestion is to delete the comparison to the one moment and two moment microphysics schemes from this paper. There is very little sensitivity and most of your points were made in the previous paper. I would focus on the difference between the 2.2 km and 12 km simulations and the current and surrogate climate.

We agree with the reviewer that figure 6 is very busy. We changed figure 6, separating cloud ice and cloud water, thereby reducing the amount of lines in this specific plot.

2. The paper treats all the results with nearly equal weight. I find it to more of a travelogue than a research paper. I think you have a message you want to convey to the reader and I would focus on that message from both the figures you show and the discussion in the text. As I mention above, I was most interested in the difference between the convective permitting and the convection parameterization simulations for both current and future climate. This message is lost in the travelogue style of presentation.

As stated above we agree with the reviewer that a stronger focus on specific results will improve the paper. As suggested above we remove the 1M vs. 2M comparison from the paper.

3. I would like to see more emphasis on the physical reason for the results.

For instance, why is the diurnal timing for convection changed going from convective permitting to convective parameterization simulations in figure 4?

This is an important research question but not the main focus of this paper. The result about the improved diurnal cycle with explicit convection is already broadly discussed in the introduction (P2, L9-18). Furthermore, we added another reference to a recent paper which addresses this question (Fosser et al. 2015, Clim. Dyn. Benefit of convection permitting climate model simulations in the representation of convective precipitation), see Page 9 Line 1. The reason for the later precipitation peak in convection-resolving simulations is at least partly understood. It is related to the representation of the life cycle of convective cells in explicit simulations, while parameterization schemes assume an ensemble of convective cells in equilibrium with the instability present in the vertical profile. The latter is particularly evident for convective adjustment schemes, which remove potential instability at every time step.

Why does the CPM simulation have less precipitation?

We do not consider these differences between CPM and CRM as being very significant, in fact we did not discuss them in the paper, although indeed CPM has slightly larger precipitation amounts. The main issue considered in the paper is the precipitation response to the two different forcing's considered (e.g. HW and VW), with HW producing a more pronounced precipitation increase. This result is not surprising, as VW addresses the case of increased vertical stratification, which will tend to suppress convection and reduce precipitation.

Is the amount it estimated close to observations?

In Fig. 11a of Keller et al. (2016), a comparison of the diurnal cycle of precipitation over Switzerland was shown for the three CTRL simulations as well as the observations. The panel shows that the diurnal cycle is better captured in the CRM simulations, and overall in reasonable agreement with observations, in particular when accounting for the observational uncertainties (discussed in Section 3.4 of Keller et al. 2016). As regards the comparison in Fig.3 of the current paper, there is again a reasonable qualitative agreement of the CTRL simulations.

Otherwise these are just model results and I haven't really learned anything other than there is a difference between the runs or not. There are only 11 days of simulation, so a focus on the physics rather than the climatology seems warranted and appropriate.

We agree that a focus on the physical interpretation makes sense. We think that the revisions have helped in bringing out the relevant points more clearly. Furthermore, there is a lot of discussion in the paper interpreting the differences between CTRL, HW and VW simulations.

**Minor comments**

Page 3, line 1. Please state the height in the atmosphere for which the north-south temperature gradient is impacted.

Corrected. It was 2m temperature.

Page 6. Line 29. Delete "steps at".

Text corrected.

Page 6. I would like to see an image of the analysis domain in this paper.

The analysis domain is shown in Fig. 1a and explained in the first paragraph of Section 2.1. It is the domain of the CRM simulations.

Page 7, line 8. I would have liked to have seen a sequence of synoptic maps characterizing the 10 day period (if nearly constant, a composite map).

Such maps can be very informative, but to keep the paper focused we would rather not increase the figure count.

Page 9. Line 22."with" should be replaced by "by".

Text corrected.

Page 10. I would like to see difference plot for figure 3. It is very difficult to see what the differences in various runs actually are and what magnitude otherwise.

Figure 3 is meant to give a qualitative feeling for the amount and distribution of accumulated precipitation during the analyzed period. For an 11 day period the influence of internal variability is rather large and difference plots are very noisy and not as informative as the area-mean precipitation changes shown in Table 2, and the mean diurnal cycles shown in Fig. 4. We do not think that too much emphasis should be attached to spatial details, as only 11 days have been considered, and as summer convection is a fundamentally chaotic process.

Page 9, line 29. Can you be more clear about the expected circulation changes?

We reformulated the whole paragraph and moved the circulation changes to the discussion section.

Page 9 Lines 5-13

Page 14, Lines 22-24

Page 11, line 9. Need to state what the differences are between the 12 km and 2 km runs and how they agree with Ban et al. (2015).

We removed this analysis from the paper see answer to major comment 2

Page 11, line 14. Need to state what the values of vertical velocity you are talking about.

We did not fully understand this comment. The distribution of vertical velocities is shown in Fig.5a.

Page 14. Lines 3-14. This is an interesting discussion of the causes for the differences in the VW and HW simulations. I would encourage a more detailed analysis as the discussion speculates more than determines what the real cause is. It might be useful to examine the evolution of clouds in detail for one or two days for both VW and HW to determine the cause. You only have 11 days, so an average does not necessarily give you a robust result.

The 11 days analyzed in this study are part of several month long free running simulations. Therefore internal variability makes it difficult to compare exactly one day of HW simulation against the same day in a VW simulation.

Page 16. Lines 15 and 16. Why is there an increase in heavy precipitation events for the CPM runs compared to the CRM runs? Do the CPM runs compare well to the CRM runs for non-heavy events?

We removed the scaling analysis from the paper, see answer to major comment 2. We removed this part of the conclusions.

Page 16, lines 24-26. I think this is the most interesting result of the paper and should be explored deeper. First of all, how does the VW change in vertical distribution physically effect the clouds? You only have 11 days of simulation, so you should be able to note some common evolution and physical changes. Second, why is the response of the CRM different than the CPM for HW and VW? This is a very interesting result that deserves more investigation.

The main difference between the CRM and CPM response to VW is the difference in upper-level clouds. This factor is now better stressed in the conclusions (see Pages 14, Liens 12-15). We think that the cloud cover changes in VW are related to reductions in convective activity. The fact that CPM is more strongly affected is difficult to interpret, as that model version is affected by a substantial overestimation of upper level clouds in CTRL.

**Final comment**

The conclusion section is much too short. There should be much deeper discussion of the results here that can help the reader understand the detailed simulation results presented in the previous section. What do you want the reader to take away from this study? I am current not sure, and that is a problem.

We agree with the comment of the reviewer. We extended and reformulated parts of our conclusion section, picking up the questions posed in the introduction.

See conclusion on Page 13-14.

[revised manuscript text omitted]